# Deciphering the Hidden Ecology and Connectivity of *Vibrio* in the Oceans

Lapo Doni [1,2,3], Joaquin Trinanes[4], Emanuele Bosi[1], Luigi Vezzulli [1,2,6] ✉ & Jaime Martinez-Urtaza [3,5,6] ✉

Long-range dispersals of marine bacteria in the oceans have remained largely indecipherable, which is particularly relevant for *Vibrio*, responsible for global epidemics in humans and animals. Here, we combine the analysis of 40 ter-abases of metagenomic data and satellite-tracked surface drifter data, from across the globe revealing that *Vibrio* are abundant members of the ocean surface and show a strong association with microplankton, which appears to govern their distribution and connectivity at a global scale. We identify long-distance biological corridors connecting *Vibrio* communities, including potentially pathogenic *Vibrio*. These corridors allow movement over thousands of kilometres in a fairly short time, with estimates of less than 1.5 years to cross an ocean basin. These findings have deep implications for the demography and community dynamics of *Vibrio* species and the epidemiology of associated diseases.

Dispersal heavily influences the geographical structure of genetic variation in bacteria. Connectivity between communities favors gene flow and limits the geographical structuring by counteracting the effects of genetic drift, thereby eroding barriers for allopatric speciation. In oceans, the absence of physical barriers for dispersal enhances interconnectivity between remote areas. However, microbial communities generally follow a distance-decay pattern[1–4], where the genetic similarity declines as distance increases. This relationship can be weaker when dispersal rates are high, for example, when currents and vectors (which in turn can be carried by currents) facilitate dispersal, while limited dispersal[5] or constraints in response to spatially structured environment[6,7] can have the opposite effect. These factors shape the biogeographical patterns and connectivity across oceans.

*Vibrio* species are natural constituents of estuarine and marine environments[8] and occupy a broad range of ocean ecosystems, from surface to deep waters[9], although their global biogeography and demography have remained poorly explored except for the main pathogenic species. Among the 150 *Vibrio* species, a dozen can cause infections in humans[10] and many more in animals[11]. *V. cholerae* and

*V. parahaemolyticus* represent the only two known marine bacterial pathogens responsible for global-scale epidemics, which are currently expanding geographically and temporally[12]. These pathogenic species are adapted to coastal conditions, and their distribution and abundance are driven by physicochemical factors, especially warm and low salinity waters. They are characterized by high genetic intra-species diversity and dynamic community structure, with pathogenic variants emerging and causing infections far away from their endemic areas, often through unknown routes and mechanisms of dispersal. Since patterns of transcontinental dispersal have been described for the two main pathogenic species[13,14], the existence of active oceanic transport of *Vibrio* communities mediated by plankton has been proposed[15–19]. The association with plankton could provide protection from the cold saline environments of the open ocean and may represent a carbon and nitrogen source for survival during prolonged journeys[20]. Long-distance dispersal may contribute to recurrent incursions of foreign communities, fostering frequent admixture processes with local communities and introducing constraints on niche specialization[20–22]. Despite such evidence, it has been conventionally assumed that *Vibrio*

[1]Department of Earth, Environmental and Life Sciences (DISTAV), University of Genoa, 16132 Genoa, Italy. [2]NBFC, National Biodiversity Future Center, 90133 Palermo, Italy. [3]Centre for Environment, Fisheries and Aquaculture Science (CEFAS), DT4 8UB Weymouth, UK. [4]Department of Electronics and Computer Science, University of Santiago de Compostela, 15706 Santiago de Compostela, Spain. [5]Department of Genetics and Microbiology, Universitat Autònoma de Barcelona (UAB), 08193 Barcelona, Spain. [6]These authors contributed equally: Luigi Vezzulli, Jaime Martinez-Urtaza ✉e-mail: luigi.vezzulli@unige.it; jaime.martinez.urtaza@uab.cat

is rarely abundant in the ocean, and when present, its scarcity prevents it from being detected in most of the metagenomic studies[23]. Consequently, the dispersal via marine transport across the oceans is difficult to prove. Plankton is transported across the world oceans by major trans-oceanic currents, which define its global-scale biogeography[24].

We hypothesized that the migratory routes defined for plankton could also act as a mechanism for the long dispersal of plankton-associated *Vibrio* across the oceans, enhancing the recurrent introductions of *Vibrio* communities originating in distant areas. These dispersal processes would link *Vibrio* (only or *Vibrio* communities) as a metacommunity, with substantial implications for their demography, population dynamics, and evolution. Here we show that, based on 40 terabases of worldwide metagenomic data and satellite-tracked surface drifter observations, *Vibrio* are abundant at the ocean surface and exhibit a strong association with plankton, whose migration routes act as biological corridors governing their distribution and connectivity at a global scale.

## Results

The TARA Oceans Expedition (2009-2013) was a global scientific research initiative aimed at exploring and understanding the diversity, distribution, and ecological roles of marine plankton across the world's oceans[25]. The project generated one of the largest shotgun metagenomic datasets from the marine environment by sampling and analyzing seawater across multiple size fractions and depths. In this study, approximately 1500 metagenomes, corresponding to around 40 terabases of FASTQ files and roughly 3 billion reads collected from 147 stations worldwide, were analyzed (Fig. S1).

### Bacterial community and *Vibrio* distribution in the oceans

Taxonomic analysis of bacterial communities, conducted using Kraken2[26] with the NCBI bacteria RefSeq database on more than 40 terabases of metagenomic data generated from across the globe, revealed distinct structural patterns in the oceans. *Candidatus Pelagibacter* showed the highest average abundance (6.4%), ranging from 18.3% in the Arctic Ocean (ARC) to 3.2% in the Mediterranean Sea (MED) (Fig. 1A). *Prochlorococcus, Synechococcus,* and *Alteromonas* showed mean abundances around 5%, with higher levels in temperate regions and lower levels in polar regions (Fig. 1A). *Vibrio* was the seventh most abundant bacterial genus, with the highest abundance (2.6%) found in the North Indian Ocean (ION) and the lowest (1.2%) in the Southern Ocean (SOC), averaging 1.8% overall (Fig. 1A). Notably, *Vibrio* was significantly more abundant in the plankton-associated fraction (PAV) (considered here purely as a size category, recognizing that it consists of a mixture of organic particles and living organisms, mainly eukaryotes) compared to the free-living fraction (FLV, 0.22–3 μm) (Wilcoxon $p < 0.001$) (Fig. 1B, see the caption for full statistics) and was more abundant in surface waters than in deeper waters (Kruskal–Wallis $p < 0.001$) (Fig. 1C, see the caption for full statistics). Refined Kraken2 classification using the Enterobase *Vibrio* database[27] retrieved approximately 160 million *Vibrio* reads. *Vibrio* alpha diversity, calculated as richness, displayed distinct latitudinal (Fig. 1D) and longitudinal (Fig. S2) patterns. Specifically, the overall richness for FLV was significantly higher than PAV (Wilcoxon $p < 0.001$). Accordingly, beta diversity analysis based on nucleotide 31-mer frequencies (k-PCoA) clearly separated FLV and PAV samples along the first axis (23.3% variation) (Fig. 1E). On the second axis (9.2% variation), FLV samples were well-defined by oceanic regions, while PAV samples exhibited overlap, suggesting a less structured biogeography and potential connectivity between regions.

Permutational multivariate analysis of variance (PERMANOVA) confirmed that both size fraction (pseudo-F $(1, 20) = 51.91$, $R^2 = 0.567$, $p = 1 \times 10^{-4}$) and, to a lesser extent, oceanic regions (pseudo-F $(4, 20) = 4.90$, $R^2 = 0.214$, $p = 3 \times 10^{-4}$) significantly influenced *Vibrio* communities. Beta diversity analysis with *Vibrio* Kraken2 taxonomy

abundances (t-PCoA) (Fig. S3) exhibited similar patterns to the k-PCoA ordination. Furthermore, the Mantel test between the two distance matrices revealed a strong and significant correlation ($r = 0.7614$, $p = 1 \times 10^{-4}$). The distinct separation between PAV and FLV may suggest important biological implications for the structure of *Vibrio* communities. Therefore, *Vibrio* species detected in the FLV and PAV fractions, as well as the biogeographical structure of each size fraction, were further evaluated.

### Vibrio species

*Vibrio* reads from both FLV and PAV, for each oceanic region, were co-assembled to achieve a higher taxonomic resolution. Co-assembly was performed by grouping stations into oceanic regions based on Longhurst Provinces and station proximity[28]. A total of 3,74,833 contigs were produced, with an average of 14,417 contigs per oceanic region and an overall average N50 of 684 bp (Supplementary Data S1). The contig taxonomy analysis identified several *Vibrio* species (Supplementary Data S2), with an overall mean confidence of 0.95 and for the potential pathogens *V. cholerae* and *V. parahaemolyticus* of 0.97 and 0.96, respectively. Interestingly, *Vibrio* species known as causal agents of both human and animal diseases generally exhibited higher abundance in the PAV (Fig. 2A, see caption for statistics details). Notably, among the 13 species significantly associated with plankton, some were potentially pathogens for humans (*V. cholerae, V. parahaemolyticus,* and *V. alginolyticus*) and for animals (*V. harveyi, V. rotiferianus, V. campbellii, V. mediterranei, V. sinaloensis,* and *V. splendidus*), while different patterns were shown for species distribution in oceanic regions (Fig. 2B). Beyond its role in the dispersal of *Vibrio* species, these findings also highlight the role of plankton in transporting potentially pathogenic species, with possible implications for both human and aquatic animal health.

### *Vibrio* biogeographical structure

*Vibrio* biogeographical structures were determined from *Vibrio* reads by analyzing the k-mer similarity values for each fraction of the surface water samples. k-mers provide unequivocal information on the occupancy of specific genomes and connectivity among communities, serving as reliable genomic tracers composed of fragments of organismal DNA (Fig. S4)[24]. For the analysis, PAVs were categorized into size groups based on their association with nanoplankton (NAV: 5–20 μm), microplankton (MiAV: 20–180 μm), and mesoplankton (MeAV: 180–2000 μm)[25]. The Bray–Curtis 31-mers matrices of each fraction were used to produce RGB-PCoAs, with station colors based on their position in the ordination. The biogeographical structures were then visualized by plotting the stations with their coordinates, while maintaining the colors from the RGB-PCoAs. As a result, the FLV fraction (Fig. 3A) exhibited the most clearly defined biogeographical patterns, consistent with the k-PCoA analysis. The nonlinear correlation coefficient (NLCC) was also employed to quantify the biogeographical structures and detect whether stations displayed spatial nonlinear patterns in each RGB-PCoA. Interestingly, as the fraction size increased, the biogeographic compartmentalization decreased, except for MiAV, which exhibited the lowest NLCC (Fig. 3B). Overall, these results suggested that plankton played a substantial role in shaping the spatial patterns of *Vibrio* communities, possibly by facilitating their large-scale displacement.

### *Vibrio* and oceanic circulation

**Vibrio transportation via ocean currents.** To investigate whether *Vibrio* displacement was related to plankton transportation via ocean currents, estimated travel time (TT) (Supplementary Data S3) and Lagrangian trajectories among stations were computed using the NOAA Global Drifter Program, which tracks ocean surface currents via a network of drifting buoys. TT has been previously proposed as a robust approach for studying dispersal mechanisms in oceans[24,29].

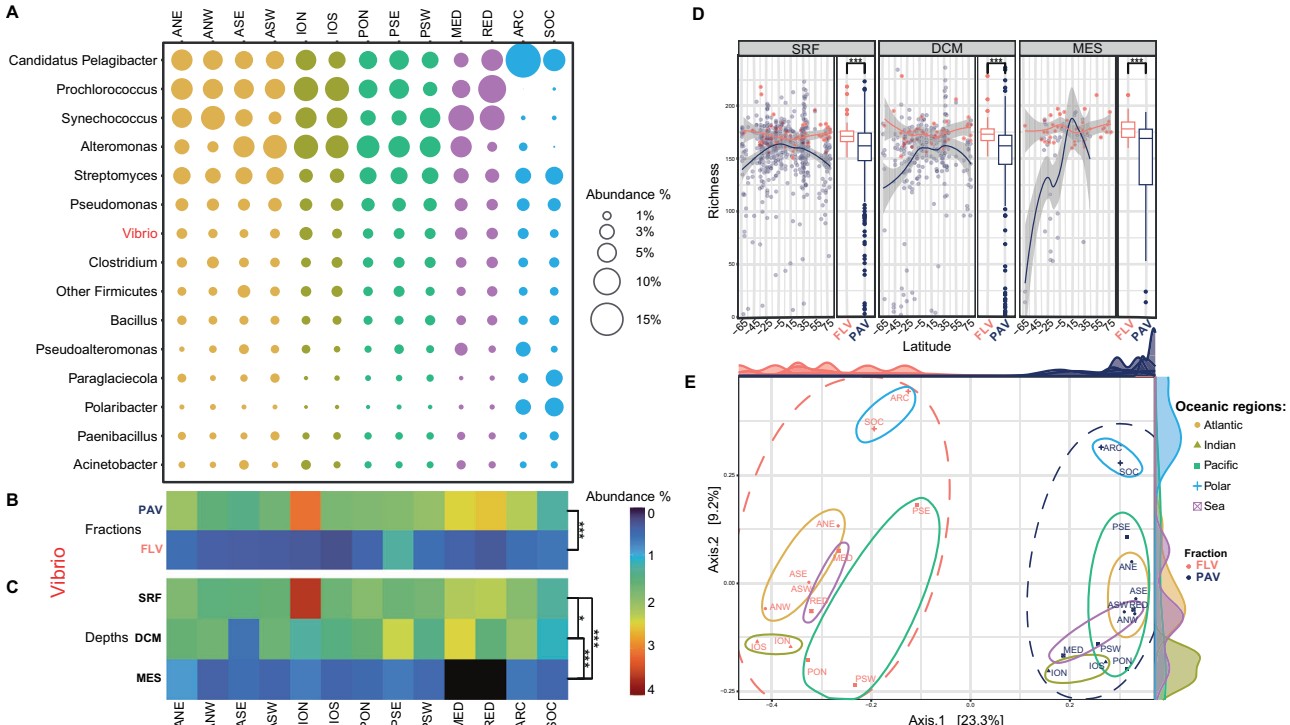

**Fig. 1 | Bacterial composition of oceanic samples. A** Bubble plot showing the top 15 bacterial genera divided for oceanic regions (columns) with *Vibrio* in the 7th position. Oceanic regions include: ANE (Northeast Atlantic Ocean), ANW (Northwest Atlantic Ocean), ASE (Southeast Atlantic Ocean), ASW (Southwest Atlantic Ocean), ION (North Indian Ocean), IOS (South Indian Ocean), PON (North Pacific Ocean), PSE (Southeast Pacific Ocean), PSW (Southwest Pacific Ocean), MED (Mediterranean Sea), RED (Red Sea), ARC (Arctic Ocean) and SOC (Southern Ocean). Heatmaps for *Vibrio* genus: **B** average of the frequencies of *Vibrio* genus in the PAV (plankton-associated fractions), which correspond to 5–20 μm, 20–180 μm, 180–2000 μm, and the FLV (free-living fraction), which corresponds to 0.22–3 μm. Statistical differences between FLV ($n = 176$) and PAV ($n = 906$) were assessed using a two-sided Wilcoxon rank-sum test ($W = 29{,}141$; $p < 2.2 \times 10^{-16}$). Asterisks in the plots indicate significance levels based on two-sided Wilcoxon rank-sum tests (*$p < 0.05$; **$p < 0.01$; ***$p < 0.001$). **C** Relative abundance of *Vibrio* genus across the depths: only metagenomes classified as SRF (Surface waters, $n = 651$), DCM (Deep Chlorophyll Maximum, $n = 325$), and MES (Mesopelagic, $n = 81$) were used for this plot. Differences across depths were firstly assessed using a Kruskal–Wallis test ($\chi^2(4) = 50.959$, $p = 2.277 \times 10^{-10}$), followed by two-sided pairwise Wilcoxon rank-sum tests with Benjamini–Hochberg correction for multiple comparison (DCM vs SRF: $W = 97502$; $p = 0.046$; MES vs SRF: $W = 13{,}413$; $p = 1.6 \times 10^{-12}$; DCM vs MES: $W = 18{,}042$; $p = 3.63 \times 10^{-7}$). Black squares mean no data. **D** *Vibrio* alpha diversity, measured as the species richness, was calculated only for samples in which *Vibrio* reads were classified at the species level, and plotted against the latitude of the sampling stations for different depths: SRF, DCM, and MES. Each point represents an independent biological replicate (one metagenomic sample) and the colors indicate size fractions: free-living Vibrio (FLV, pink) and plankton-associated Vibrio (PAV, blue). The plot includes LOESS curves (shaded bands = 95% confidence interval), boxplots summarize richness values for each size fraction and depth: the center line represents the median, box bounds indicate the interquartile range (IQR) defined as the difference between the first quartile (Q1) and the third (Q3), whiskers extend to the minimum and maximum values within [Q1 − 1.5 × IQR, Q3 + 1.5 × IQR], horizontal lines represent median values. Sample sizes were FLV ($n = 81$) and PAV ($n = 550$) for SRF, FLV ($n = 49$) and PAV ($n = 267$) for DCM, and FLV ($n = 39$) and PAV ($n = 42$) for MES. Differences in alpha diversity between size fractions were assessed separately for each depth using two-sided Wilcoxon rank-sum tests with Benjamini–Hochberg correction for multiple comparisons (SRF: $W = 30761$; $p = 9.03 \times 10^{-8}$, effect size = 0.381, 95% Confidence Intervals = 0.260–0.490; DCM: $W = 9698$; $p = 1.19 \times 10^{-7}$, effect size = 0.482, 95% Confidence Intervals = 0.336–0.606; MES: $W = 1152$; $p = 0.00169$, effect size = 0.406, 95% Confidence Intervals = 0.176–0.594). **E** 31-mers based PCoA (k-PCoA) of the *Vibrio* across the samples, (colors indicate fractions and oceanic regions); the percentage of the variation explained by each axis is indicated in parentheses after the axis label.

Similarities (Supplementary Data S4–S7) derived from the 31-mers Bray–Curtis dissimilarity matrices of samples collected from surface waters for each size fraction (FLV, NAV, MiAV, and MeAV), were compared with TT by calculating cumulative Spearman correlations (Fig. 4A). Notably, maximum cumulative correlation values were observed for pairs of stations separated by a TT of up to 1.5 years across all size fractions (Fig. 4A, peak of yellow-dashed line) which aligns with a previous study examining plankton TT from TARA samples[24]. Consequently, this threshold was selected for further analyses (Fig. 4B). For each size fraction, *Vibrio* sequence k-mers similarity was correlated with TT up to 1.5 years. The same approach was applied to correlate pairwise similarity with distance up to 5000 km, a spatial scale representative of the distance to cross an oceanic region[24]. Two-sided Fisher's z-test was used to determine whether these correlations differed among fractions (detailed statistics about correlations and Fisher's z-test for TT in Supplementary Data S8 and geographical

distance in Supplementary Data S9). The results showed that correlations between similarity and TT significantly differed when comparing only FLV to all other size fractions. For geographic distance, significant differences in correlation were found between FLV and the other fractions, but not between MiAV and MeAV. When comparing the slopes of the *Vibrio* similarity patterns in relation to TT and geographic distance (Fig. 4C, detailed statistics in Supplementary Data S10), FLV showed clear negative slopes for both descriptors, consistent with a stronger spatial structuring and dispersal limitation. In contrast, for MiAV and MeAV, the slopes related to geographic distance were close to zero, indicating a negligible relationship, while the slopes for TT were less negative compared to FLV, suggesting a higher degree of dispersal across locations through plankton transported by oceanic currents. NAV exhibited an intermediate pattern between higher plankton size and FLV. Notably, a higher genetic similarity was observed in the PAV fraction (especially MiAV) compared to FLV across

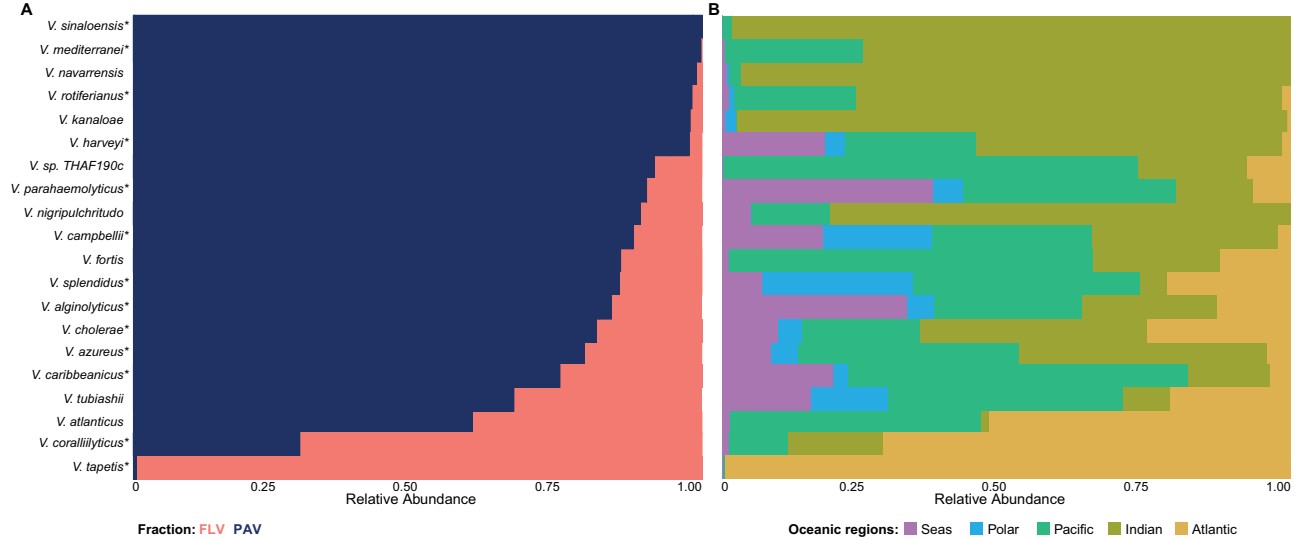

**Fig. 2 | Vibrio species distribution in the Oceans.** Top 20 *Vibrio* species distribution. **A** Distribution of Vibrio species between the free-living (FLV) and plankton-associated (PAV) fractions based on their co-assembly for each oceanic region (FLV $n = 13$; PAV $n = 13$). Species with significantly different abundance between fractions were identified using two-sided pairwise Dunn tests with Bonferroni correction for multiple comparisons across the top 20 species, and significant differences are indicated with * ($p < 0.05$). Significant species included: *V.*

*alginolyticus* ($p = 0.00054$), *V. harveyi* ($p = 0.00301$), *V. cholerae* ($p = 0.00365$), *V. parahaemolyticus* ($p = 0.00977$), *V. rotiferianus* ($p = 0.01028$), *V. azureus* ($p = 0.01541$), *V. campbellii* ($p = 0.01582$), *V. mediterranei* ($p = 0.02390$), *V. coralliilyticus* ($p = 0.02484$), *V. tapetis* ($p = 0.02803$), *V. sinaloensis* ($p = 0.03579$), *V. splendidus* ($p = 0.03969$), and *V. caribbeanicus* ($p = 0.04310$). **B** Distribution of *Vibrio* species among oceans and seas (Mediterranean Sea and Red Sea).

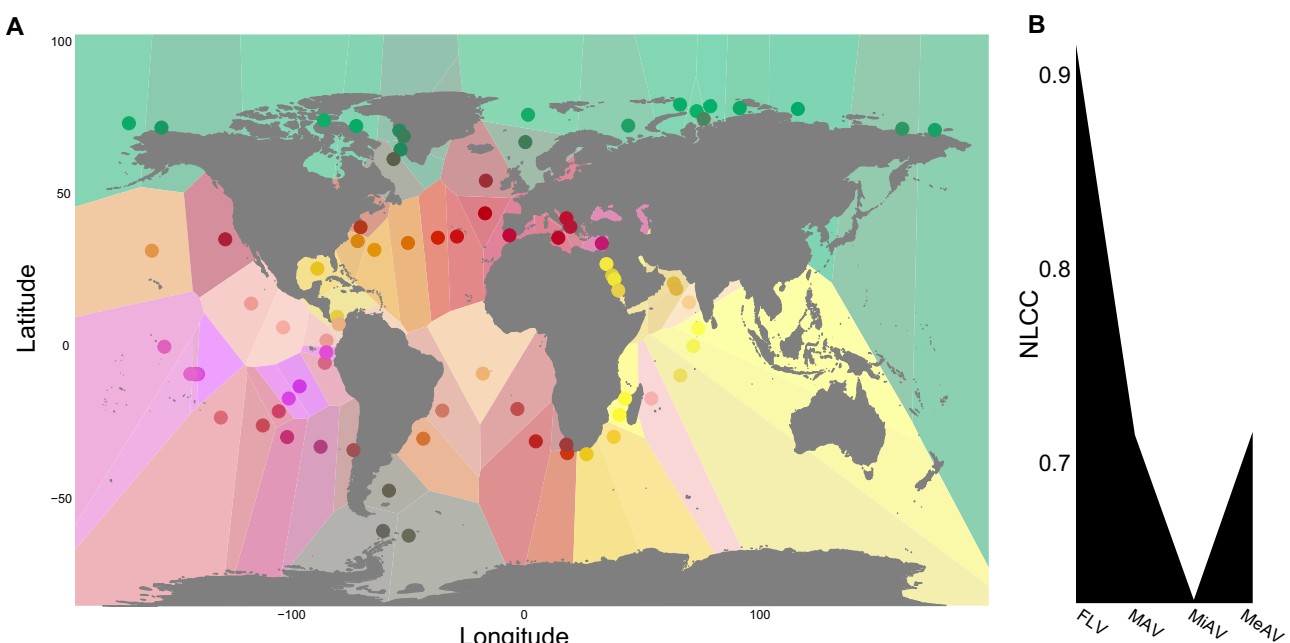

**Fig. 3 | Biogeographical structures of Vibrio in the Oceans.** *Vibrio* biogeographical structures of surface stations, for the FLV (0.22–3 μm) fraction **A**, colors of the stations are derived from the RGB-PCoA based on *Vibrio* metagenomic

dissimilarities; similar colors indicate similar communities. **B** The nonlinear correlation coefficient (NLCC) was calculated for each size fraction based on the differential patterns in the RGB-PCoA stations.

the ocean (Fig. 4D). In addition, MiAV showed a significantly higher similarity associated with TT than with geographic distance (see Fig. 4D for full statistical details). These findings are consistent with the dispersion of large plankton fractions across oceanic regions, which is mainly driven by oceanic currents (TT), rather than shaped by geographical distance[24]. In contrast, the dispersion of FLV is less affected by currents for transportation. The comparison of similarities for MiAV and MeAV between TT shorter and longer than 1.5 years (Fig. S5)

showed significantly higher similarity values for TT < 1.5 years (see Fig. S5 caption for full statistical details).

**Connectivity of Vibrio communities at a global oceanic scale.** Network analysis was performed to further assess the connectivity among *Vibrio* communities, focusing on pairs of stations (nodes) connected by TT of less than 1.5 years, with edges weighted according to their sequence similarities. Accordingly, MiAV network

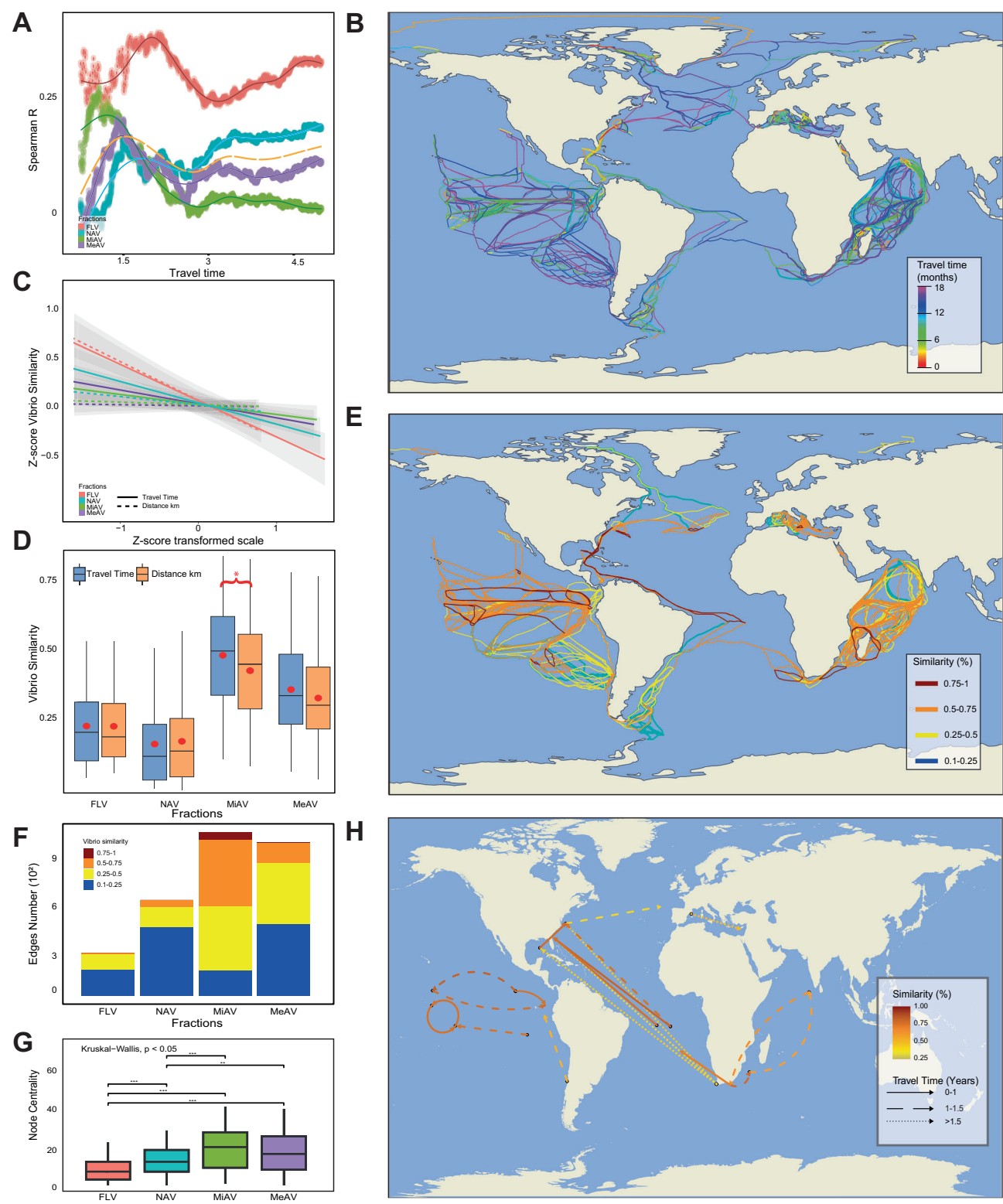

(Fig. 4E) had the highest number and strength of edges, compared to the FLV network, which exhibited fewer and weaker edges compared to the other fractions (Fig.4F, Fig. S6, Movie S1). In particular, compared to FLV, MiAV showed a decrease in the weak similarity range edges (<0.25) by 3.87%, while medium-weak (0.25 < 0.5) and medium-strong (0.5 < 0.75) similarities (edges) increased by 316.04% and 5675.00%, respectively. Notably, the highest values of sequence similarity (0.75–1) were observed in the larger PAV

fractions (MiAV and MeAV). Partial Mantel test comparing *Vibrio* composition for each fraction with surface water temperature and salinity showed that these environmental factors had the strongest effect on FLV ($R = 0.8$, $p = 1e-04$) compared to NAV ($R = 0.2$, $p = 1e-04$), MiAV ($R = 0.5$, $p = 1e-04$), and MeAV ($R = 0.1$, $p = 6e-04$). These results suggest that FLV might exhibit adaptation to local environmental conditions, likely due to isolation and limited dispersal, which may confer a degree of ecological cohesion.

**Fig. 4 | *Vibrio* surface dispersion through biological corridors in the Oceans.**
**A** Spearman cumulative correlation between *Vibrio* diversity and travel time (TT) for each different size fraction. The yellow-dashed line represents the overall trend.
**B** Lagrangian trajectories corresponding to the minimum travel time, referred to as colors of the edges, among all stations connected by a maximum of 1.5 years.
**C** Distance and travel time decays based on the regression of Z-score transformed *Vibrio* similarity plotted against Z-score of logarithmically transformed distance in km (dashed line) and travel time (continuous line). Geographic distance was calculated as the shortest path between stations, avoiding land crossings. Shaded bands represent 95% confidence intervals of the fitted linear trends. Detailed statistics of Spearman correlations, two-sided Fisher's z-tests (TT: Supplementary Data S8, distance: Supplementary Data S9), and linear regressions (Supplementary Data S10) are available in the supplementary data. **D** Boxplot showing the comparison for each fraction (FLV: 0.22–3 µm, NAV: 5–20 µm, MiAV: 20–180 µm, and MeAV: 180–2000 µm), of *Vibrio* similarity values with travel time (blue) and distance in kilometers (orange) without crossing lands. Boxplot bounds indicate the interquartile range (IQR) defined as the difference between the first quartile (Q1) and the third (Q3), whiskers extend to the minimum and maximum values within [Q1 − 1.5× IQR, Q3 + 1.5× IQR], and horizontal lines represent median values. The red dots represent the average value. Sample sizes are $n = 148$ (FLV), 309 (NAV), 524 (MiAV), and 488 (MeAV) for TT and $n = 448, 832, 1245$, and 1282 for distance in km, respectively. Statistical differences between travel time and geographic distance

within each fraction were assessed using two-sided Wilcoxon rank-sum tests. Only the MiAV fraction showed a significant difference ($W = 347672$, $p = 0.0285$, *).
**E** Network among stations connected by a travel time less than 1.5 years for the MiAV fraction (20–180 µm), connections are color-coded based on the similarity of the *Vibrio* communities between stations. **F** Frequency of edges and their strength of similarity for each fraction in their corresponding network. **G** Node centrality values for each size fraction in their corresponding network. Box plots summarize node centrality for each fraction: Boxplot bounds indicate the interquartile range (IQR) defined as the difference between the first quartile (Q1) and the third (Q3), whiskers extend to the minimum and maximum values within [Q1 − 1.5× IQR, Q3 + 1.5× IQR], and horizontal lines represent median values. Statistical differences in node centrality among size fractions were first assessed using a Kruskal−Wallis test ($\chi^2$ (3) = 55.8, $p = 4.68 \times 10^{-12}$), followed by two-sided pairwise Wilcoxon rank-sum tests with Benjamini−Hochberg correction. FLV differed significantly from NAV ($n = 66$ vs 95, $W = 1910$, $p = 5.04 \times 10^{-5}$, ***), MiAV ($n = 66$ vs 114, $W = 1540$, $p = 2.43 \times 10^{-10}$, ***) and MeAV ($n = 66$ vs 120, $W = 1929$, $p = 2.16 \times 10^{-8}$, ***). The NAV fraction also differed significantly from MiAV ($n = 95$ vs 114, $W = 3688$, $p = 1.08 \times 10^{-4}$, ***) and MeAV ($n = 95$ vs 120, $W = 4581$, $p = 0.016$, whereas the comparison between MiAV and MeAV was not significant ($n = 114$ vs 120, $W = 7632$, $p = 0.126$). **H** Long-distance biological corridors for MiAV, identified as connections between stations over long distances, with high similarity and short TT, spanning one or more oceanic regions.

Stations with high node centrality serve as stepping-stones to connect *Vibrio* communities. Notably, node centrality values (degree), representing the connectivity of each station within the network, were significantly higher for PAV (Fig. 4G, see caption for full statistics). Regions of high centrality identified stations with a higher importance as hubs for genetic connectivity[30], which may facilitate gene flow and reduce genetic differentiation[31]. Most of these stations were located in the Indian and Pacific Oceans (Fig. S7A, B), playing a crucial role in facilitating large-scale dispersal of *Vibrio* communities throughout the marine environment. The MiAV high-connectivity hubs closely align with previously described transcontinental transmission routes for pathogenic *Vibrio*[13]. Additionally, long-distance biological corridors for MiAV (Fig. 4H, Supplementary Data S11) were identified as connections between stations over long distances, with high similarity and short TT, spanning one or more oceanic regions. These corridors revealed interesting patterns of *Vibrio* circulation in major hubs like the Indian and Pacific Oceans. For example, between the Indian Ocean (ION) and the South Indian Ocean (IOS), the estimated TT was 1.1 years, with a similarity of 0.7. In the Pacific Ocean, a connection from the south to the equator had a TT of 1.3 years and a similarity of 0.5. Other circulations within the Pacific Ocean included one linking the Southeast Pacific (PSE) and Southwest Pacific (PSW) Oceans with a similarity of 0.8 and a TT ranging from 0.9 to 1.2 years, and another connecting the North Pacific Ocean (PON) to PSE with a similarity of 0.8 and a TT ranging from 0.6 to 1.4 years. Detailed analysis of the MiAV average TT and similarities for intra- and inter-oceanic regions is shown in Fig. S8. Notably, high average similarity was observed between the Northwest Atlantic (ANW) and Northeast Atlantic (ANE) Oceans, with an average of 0.7 in both directions and an average TT of 1.31 years from West to East and 1.25 years in the opposite direction. From Southeast Atlantic (ASE) to Southwest Atlantic (ASW), the average similarity was 0.71, with a TT of 0.95 years. Among intra-oceanic regions, ASE exhibited the highest similarity of 0.65, with an average TT of 0.69 years.

## Discussion

The present study challenges the conventional assumption that the open ocean is a hostile environment to most *Vibrio* species, revealing them as natural inhabitants of the pelagic ecosystems. *Vibrio* communities prevailed in the upper ocean layers and, although they could be identified as part of the free-living community, they were predominantly associated with plankton. FLV exhibited niche partitioning and a biogeographical structure, likely shaped by limited dispersal and

isolation, which may contribute to ecological cohesion. These results are consistent with previous studies, suggesting that microbes evolve faster in the absence of effective dispersal than they are diluted by ocean currents, leading to distinctive biogeographic patterns[32]. Microbial communities with a lower dispersal may have higher richness[33], as observed for FLV compared with the PAV. The identification of a strong negative relationship between FLV similarity and TT and distance supports this evolutionary context, where neutral evolution and genetic drift may sustain the observed geographical distribution and biodiversity profiles. On the other hand, environmental factors appear to play a substantial role in shaping the genetic diversity and similarity in communities with low dispersal ability (FLV) compared to those undergoing long-range dispersal (PAV), which is consistent with previous studies supporting the role of the dispersal ability in modulating the relevance of environmental factors[6,7]. In contrast, PAV, representing the majority of *Vibrio* in the oceans, showed a different community structure and dynamics, with a more diffuse biogeographical distribution and weaker similarity decay with TT, which may support the existence of active dispersal across oceanic regions. It can be argued that the higher genetic similarity of PAV communities across distant oceanic regions results from interacting mechanisms such as plankton host selection and oceanic transport. While host selection imposes adaptive constraints favoring communities suited to specific plankton hosts, ocean currents connect communities over time, enabling the recurrent introduction of PAV. The stronger correlation observed between genetic similarity and TT in the PAV fraction, as compared to geographic distance, along with the higher similarity among MiAV and MeAV communities connected by TT within 1.5 years compared to those connected by longer TT, strongly supports oceanic transport via currents as the primary driver of large PAV dispersal across the oceans. However, plankton-mediated selection cannot be entirely excluded, as it represents a biologically relevant ecological interaction. Unlike FLV, dispersal of PAV was more influenced by oceanic transport than by geographic distance. Thus, PAV and FLV communities may evolve under different constraints[34]. While there were evident differences between FLV and both MiAV and MeAV patterns, the NAV fraction was like a transitional group between these two major patterns, in line with previous observations for the Vibrionaceae family[35].

The connectivity observed for *Vibrio* communities was consistent with the transoceanic plankton migration routes connecting distant regions[24], supporting our hypothesis that these migratory routes could

also serve as a mechanism for the long-distance dispersal of plankton-associated *Vibrio* across the oceans. The present study provides robust evidence of a plankton size-dependent effect on the interconnectivity of *Vibrio* communities globally, following the dominant trajectories of the oceanic currents and plankton migrations. Because TT-based trajectories correspond to the dominant oceanic pathways, which transport the largest volume of ocean water and its contents, such as plankton, nutrients, and heat. Indeed, TT is not only considered a proxy for the time required for the displacement in the ocean, but also for the biotic interactions of what is transported within the currents[24]. The effective oceanic transport mechanisms have also been supported by the absence of an isolation-by-distance pattern and weak biogeographical and community structure, consequences of the lack of genetic barriers to exchange between communities. The South Pacific and Indian oceans act as major hubs for the *Vibrio* distribution. Notably, these regions are considered the major hotspots for *Vibrio* diseases in the world[13,14], with frequent transcontinental epidemic spread events following similar dispersal trajectories to those described in this study. These observations are also consistent with results obtained from the analysis of the global collection of *V. parahaemolyticus*, which concluded that the high diversity and low differentiation of these communities resulted from the long-range dispersals and local admixing of communities[36]. The ocean circulation delineated in this study ensures the migration and dispersal of *Vibrio* at a global scale following the major long-distance biological corridors in the oceans.

This recurrent admixture of introduced *Vibrio* assemblages from distant locations with local communities over a relatively short period, estimated at less than 1.5 years, may be responsible for the establishment of environmental reservoirs, potentially leading to the emergence of pathogenic variants and disease outbreaks under favorable conditions. This is corroborated by the frequent colonization events, documented for *Vibrio* globally[14,36], which counteract the biogeographic effects on community structure, imposed by drift and/or selection, eroding the distance-decay patterns. This work emphasizes the importance of plankton acting as a driver for the displacement of *Vibrio* and potential human pathogens, such as *V. parahaemolyticus*, *V. cholerae*, and *V. alginolyticus*, across oceans.

The TARA Oceans dataset is currently the largest and most comprehensive marine metagenomic dataset produced to date[25]. Nevertheless, potential caveats may include sampling constraints, particularly the effects of seasonality or interannual variation on shaping *Vibrio* communities and their associations with plankton, although these are not expected to substantially influence the main findings of this study. Such limitations should be addressed in future studies, along with the investigation of *Vibrio*-plankton interactions to potentially uncover species- or strain-specific associations and identify plankton vectors that may contribute to the long-range dispersal of pathogenic *Vibrio* strains and the epidemiology of *Vibrio* diseases. The role of climate change in these interactions is also a timely topic that warrants further investigation. Indeed, *Vibrio* is considered one of the most responsive organisms to climate change-induced conditions in the ocean[12,37,38]. The increasing warming trends are providing suitable ecological conditions for the *Vibrio* colonization of a growing number of regions worldwide[39]. This favors the introduced communities for a successful establishment and the subsequent incorporation in the local communities[20,36,40], as shown by the poleward expansion of *Vibrio* infections in the Northern Hemisphere[38,39,41]. However, climate change is affecting the oceans in multiple ways, including altering the speed and routes of the major ocean currents[42] and altering the biogeographical patterns of plankton distributions[43]. As shown in this study, both aspects are intrinsically linked to the dispersal dynamics and connectivity of *Vibrio* communities and govern the microbial community assembly across spatial scales. In consequence, we foresee substantial changes in the global demography of *Vibrio* communities in the future, resulting from the conditions imposed by climate

change, with collateral effects on the epidemiology of *Vibrio*-related diseases.

## Methods

### TARA metagenomes

The general aim of the TARA Oceans expedition was to assess the complexity of ocean life across comprehensive taxonomic and spatial scales, sampling the oceans worldwide following standardized protocols for the collection and for data production[25]. The methodology used for the sampling, size fractionation, DNA extraction, and shotgun metagenomic sequencing has already been extensively described[25,44,45]. However, for each station, usually water from three different depths was sampled: surface, deep chlorophyll maximum, and mesopelagic waters. Then, the samples were serially filtered, with a prokaryote-enriched fraction (FLV: 0.22–3 µm) and three eukaryote-enriched fractions (NAV: 5–20 µm, MiAV: 20–180 µm, and MeAV: 180–2000 µm). The abbreviations refer to the *Vibrio* reads obtained from these fractions. Moreover, the PAV fraction, which corresponds to the three plankton size categories, is recognized as a mixture of organic particles and living organisms (e.g., a mix of eukaryotes and particle-associated prokaryotes). In this study, all available shotgun metagenomes ($n = 1485$) belonging to the four NCBI TARA Oceans project (PRJEB1787, PRJEB4352, PRJEB9740, and PRJEB9691) were downloaded and analyzed. Data corresponded to approximately 40,000 Gbases of fastq files for a total of 3 billion reads[46]. The codes used for the analysis of the downloaded metagenomes are available at https://github.com/LDoni/Deciphering-the-Hidden-Ecology-of-Vibrio-in-the-Oceans, and the pipeline workflow is briefly shown in Fig. S9.

### Metagenomic reads analysis

Metagenomic reads were quality checked with trimgalore v0.6.7. The analysis of the bacterial composition was conducted with Kraken2 v2.1.2[26] using the RefSeq NCBI Bacteria database (downloaded in 2021). Metagenomic reads corresponding to the same fraction, depth, and station were aggregated together. The frequencies of bacterial genera were normalized using Bracken v2.6.2[47]. The top bacterial genera in oceans were visualized as a bubble plot[46]. *Vibrio* genus average abundances across different fractions and depths were highlighted in the heatmaps. Both plots were generated using the R package GGplot2[48]. Global maps were made with Natural Earth (naturalearthdata.com). Metagenomic reads classified as *Vibrio* were extracted using the extract_kraken_reads.py tool from KrakenTools[49]. These reads were subsequently reclassified using a custom Kraken2 database, generated with the *Vibrio* genomes present in The *Vibrio* database from Enterobase in 2021 (available at: https://enterobase.warwick.ac.uk/species/index/vibrio[27]), following the Kraken2 manual. The frequencies of *Vibrio* species were normalized using Bracken[47]. The subsequent analyses were performed in R using the Phyloseq package[50] for analysis and GGplot2 for the visualizations[48]. The *Vibrio* alpha diversity, estimated as richness, was calculated using *Vibrio* species frequencies transformed into presence–absence and then plotted against latitude and longitude for each specific depth. Beta diversity was visualized as a PCoA ordination based on the Bray–Curtis dissimilarity matrix obtained from the *Vibrio* species abundances (t-PCoA). Beta diversity based on Bray-Curtis dissimilarity matrix was also assessed using k-mers (k-PCoA) calculated with Simka[51]. *Vibrio* k-mer frequencies were calculated using nucleotide 31-mers following normalization of the number of reads and filtering out low complexity regions. Specifically, highly conserved multicopy genes, such as ribosomal proteins, rRNA, tRNAs, IS elements, and other related components in prokaryotic genomes, were filtered out (read-shannon-index set to 1.5) before the analysis. To help the interpretation of *Vibrio* k-mer similarity patterns, simulations were performed on synthetic metagenomes by varying the number of shared strains, k-mer size, strain divergence, and sequencing depth. Briefly: five synthetic metagenomes, each composed of 30

*Vibrio* reference genomes from NCBI RefSeq and differing in their percentage of genome (strain) sharing, were generated using MESS. Subsequently, different k-mer lengths (e.g., 21, 31, 51) were used to assess the relationship between k-mer similarity and the known number of shared species in the synthetic metagenomes, using Simka (https://github.com/GATB/simka). To test the effects of mutations on k-mer similarity, an in-house Python script was used to introduce random mutations into the metagenomic reads by altering bases at a specified percentage. To evaluate the impact of comparing metagenomes with different numbers of reads, subsampled metagenomes were generated using SeqKit (https://github.com/annalam/seqkit). PERMANOVA with 9999 permutations from adonis2 of the Vegan package[52] was used to compute the model to evaluate the effects of the fractions and the oceanic regions on the Bray–Curtis dissimilarity matrix used for the k-PCoA. Metagenomic dissimilarity based on nucleotide 31-mers has been shown to be genome/species specific[53] and a reliable k-mer size to evaluate differences in the genomic identity of organisms[24]. To assess the correlation between the two Bray-Curtis dissimilarity matrices (based on taxonomy counts and k-mers), a Mantel test was performed in R using the Mantel function of the Vegan package[46,52].

### *Vibrio* species analysis

Metagenomic *Vibrio* reads were co-assembled for each oceanic region, using geographically related stations, following the approach previously used for other TARA Oceans metagenomic co-assemblies[28] using megahit[54] for both FLV and PAV. Taxonomy classification was assigned with Contig Annotation Tool (CAT)[55] using its preconstructed NCBI nr database (v.2021-01-07)[46]. The relative abundance of *Vibrio* species was calculated as TPM, mapping *Vibrio* reads on the *Vibrio* contigs using Salmon[56] with the metagenomic flag, for FLV and PAV, for each oceanic region, and was normalized with the relative log expression method[57] using MicrobiomeMarker R package (available at: https://github.com/yiluheihei/microbiomeMarker). The distribution of the top 20 *Vibrio* species was visualized between fractions and among oceanic regions. Pairwise Wilcoxon test was used to test significant differences of species distribution between FLV and PAV[46].

### *Vibrio* biogeographical structure

*Vibrio* biogeographical structures were evaluated for each size fraction (i.e., 0.22–3 μm, 5–20 μm, 20–180 μm, and 180–2000 μm) based on the PCoA ordination of the Bray–Curtis 31-mers dissimilarity calculated with Simka, from samples collected only in the surface waters and filtering low complexity reads[46]. Thus, an RGB-PCoA was produced by assigning to each station a color based on its position in the ordination, taking account of the axis variation and the spatial position of the first three axes to map the stations into RGB color values. Thus, stations with similar communities shared the same range of colors. The RGB-PCoA were used to calculate the heuristic nonlinear correlation coefficient (NLCC) using the 'nlcor' R package (available at: https://github.com/ProcessMiner/nlcor), which can detect nonlinear pattern on the distribution of the stations composing the RGB-PCoA. The heuristic method involves adaptively identifying multiple local regions of linear correlations to estimate the overall nonlinear correlation. Finally, the stations were plotted on a global map, made with Natural Earth, based on their geographical coordinates, with each station colored using the relative RGB color[46].

### *Vibrio* and oceanic circulation

In order to assess the oceanic connectivity among stations, the DrifMLP package[58] was used. The Six-hourly Interpolate Database[59] (https://www.aoml.noaa.gov/phod/gdp/interpolated/data/all.php) of the Global Drifter Program, which comprises observations of ocean surface currents from a network of drifting buoys, was used as input to estimate the connectivity between stations. By tracking the Lagrangian trajectories of drifters among stations, the potential pathways were obtained, revealing their connectivity patterns and travel time[46]. Following the package recommendations, the bootstrapping and grid rotation to estimate the uncertainty in the trajectory and travel time estimates were applied. Each *Vibrio* Bray-Curtis 31-mers dissimilarity matrix, divided by size fractions (i.e., FLV 0.22–3 μm, NAV 5–20 μm, MiAV 20–180 μm, and MeAV 180–2000 μm) selected from samples collected in the surface waters, was used to compute the cumulative correlations between *Vibrio* k-mers diversity and travel time using the Spearman rank correlation coefficient. Based on the result of this analysis and previous results from analogous analyses using TARA samples[24] and from an ecological perspective, we determined a travel time cutoff of 1.5 years between stations[46]. The travel time of 1.5 years corresponds to the time needed to travel across an oceanic basin or gyre[60]. Subsequently, Spearman correlation using the cor function of the stats R package[61] between *Vibrio* k-mer similarity and travel time up to 1.5 years, and the Spearman correlation between *Vibrio* k-mer similarity and the distance within 5000 kilometers (which is on the order of the distance to cross an oceanic gyre[24]) were calculated[46]. Distance in km among stations was calculated using searoute.py (available at: https://github.com/genthalili/searoute-py), a Python package for generating the shortest sea route between two points, avoiding land using the Haversine formula[46]. Fisher z-test from the diffcor R package[62] was applied to determine if there were significant differences among the correlations of the four fractions in both Spearman correlation analyses. Moreover, to compare the *Vibrio* similarity slopes, linear models of z-score *Vibrio* similarity and z-score of log-transformed travel time and distance in km were obtained. Comparisons among all fractions divided by TT and distance in km were visualized using boxplots and tested for significant differences with the Wilcoxon test. Differences in similarity between MiAV and MeAV communities connected by <1.5 years or >1.5 years were visualized using boxplots and tested for statistical significance with the Wilcoxon test. The network approach was used to investigate the *Vibrio* k-mers metagenomic similarity in relation to travel time, for each fraction, removing outliers. The plot was generated on a global map, made with Natural Earth, using Python Cartopy (available at: https://pypi.org/project/Cartopy/), displaying the trajectories of the *Vibrio* similarities connected by station separated up to a 1.5 years of travel time[46]. The longitudes were adjusted to ensure a smooth transition across the antimeridian. The 4-panel video displaying the trajectories for each fraction was produced following similar preprocessing steps used for the static networks. The video was created by updating a set of frames with the trajectories and adjusting the time indicator as the animation advanced and then was saved as an MP4 file with a frame rate of 20 frames per second. *Vibrio* similarities for each fraction were divided into ranges: weak (0.1–0.25), weak-medium (0.25–0.5), medium-strong (0.5–0.75), and strong (0.75–1), and summarized for each fraction as a bar plot[46]. Subsequently, to estimate the influence of local conditions on *Vibrio* communities, a partial mantel test using the Vegan package was applied to correlate with Spearman the k-mers Bray-Curtis distance matrix for each fraction to the Euclidean matrix based on the surface water temperature and salinity, as proxies of local conditions, removing the effect of spatial autocorrelation with the Euclidean distance matrix of geographical coordinates of the sampling stations. Temperature and salinity were recovered from the environmental context of all samples from the TARA Oceans Expedition metadata[63] (available at: https://doi.pangaea.de/10.1594/PANGAEA.875579). Network theory metrics were applied to calculate measures of node centrality using the igraph R package[64]. Statistics of the distribution of node centrality among fractions were computed using Kruskal–Wallis and Wilcoxon tests and plotted with Ggpubr (available at: https://github.com/kassambara/ggpubr).

## Reporting summary

Further information on research design is available in the Nature Portfolio Reporting Summary linked to this article.

## Data availability

The metagenomic data (*n* = 1485) used in this study are available in the NCBI TARA Oceans BioProject database under accession codes PRJEB1787 [www.ncbi.nlm.nih.gov/bioproject/196960], PRJEB4352 [https://www.ncbi.nlm.nih.gov/bioproject/213098], PRJEB9740 [https://www.ncbi.nlm.nih.gov/bioproject/288558], and PRJEB9691 [https://www.ncbi.nlm.nih.gov/bioproject/287904]. Temperature and salinity values for each sample were retrieved from the TARA Oceans expedition metadata, available at PANGAEA under accession 875579. The *Vibrio* genomes from the EnteroBase database used as input for kraken2 classification are available at EnteroBase[27] [https://enterobase.warwick.ac.uk/species/index/vibrio]. The Six-hourly Interpolate Database of the Global Drifter Program used as input to estimate the travel time between stations is available at NOAA AOML[59] [https://www.aoml.noaa.gov/phod/gdp/interpolated/data/all.php].

## Code availability

The codes used for the analysis and the plots are available at: https://github.com/LDoni/Deciphering-the-Hidden-Ecology-of-Vibrio-in-the-Oceans and deposited in the Zenodo repository: https://doi.org/10.5281/zenodo.14677762.

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

## Acknowledgements

We thank the following institutions for financial support: National Recovery and Resilience Plan (NRRP), Mission 4 Component 2 Investment 1.4—Call for tender No. 3138 of 16 December 2021, rectified by Decree n.3175 of 18 December 2021 of Italian Ministry of University and Research funded by the European Union—NextGenerationEU; Award Number: Project code CN_00000033, Concession Decree No. 1034 of 17 June 2022 adopted by the Italian Ministry of University and Research, CUP D33C22000960007, to L.D., E.B. and L.V.; Center for Environment, Fisheries and Aquaculture Science (CEFAS) to L.D.; National Oceanic and Atmospheric Administration (NOAA) Atlantic Oceanographic and Meteorological Laboratory, NOAA CoastWatch and OceanWatch to J.T.; Spanish Ministry of Science and Innovation (PID2021-127107NB-I00 and PID2024-159955NB-100) to J.M.U. This study is a contribution to the project "National Biodiversity Future Center NBFC", funded under the National Recovery and Resilience Plan (NRRP), Mission 4 Component 2 Investment 1.4 Call for tender No. 3138 of 16 December 2021, rectified by Decree n.3175 of 18 December 2021 of Italian Ministry of University and Research funded by the European Union Next Generation EU; Award Number: Project code CN_00000033, Concession Decree No. 1034 of 17 June 2022 adopted by the Italian Ministry of University and Research, CUP D33C22000960007. We would also like to thank the Tara Oceans Foundation for providing the metagenomic data and the Center for Environment, Fisheries and Aquaculture Science (CEFAS) for the support provided.

## Author contributions

L.D., L.V. and J.M.U. contributed to the conceptualization of the study. The methodology was developed by L.D., J.T., E.B., L.V. and J.M.U. The investigation was carried out by L.D., J.T., E.B., L.V. and J.M.U. Visualization was performed by L.D., J.T. and J.M.U. Funding was acquired by J.T., L.V. and J.M.U. Project administration was conducted by L.V. and J.M.U. Supervision was provided by L.V. and J.M.U. The original draft of the manuscript was written by L.D., L.V. and J.M.U.

## Competing interests

The authors declare no competing interests.
