## [Transparent Peer Review File · Nature Communications]

Deciphering the Hidden Ecology and Connectivity of Vibrio in the Oceans

Corresponding Author: Professor Jaime Martinez-Urtaza

Version 0:

Reviewer comments:

Reviewer #1

(Remarks to the Author)

Overview

This manuscript describes the Vibrio community profile and dispersal across the oceans. In short, the authors describe the abundance of Vibrio in ocean surface waters, the deep chlorophyll maximum, and the mesopelagic, with an emphasis on the surface waters for more in-depth analyses. This study focused on broad trends and dispersal of Vibrio across the ocean based on previously established models. The authors highlight dispersal patterns amongst the plankton-associated Vibrio and oceanic currents, focusing on those areas with a transit time of 1.5 years. The key finding in this study highlights the plankton-associated Vibrio communities within different size fractions, and demonstrates clear evidence of the plankton size-dependent effect on the interconnectivity of Vibrio communities globally.

An area in which this manuscript that could be improved upon is the discussion of the differences between the free-living Vibrio and the plankton-associated Vibrio fractions globally. The analysis between these two is surface-level (in particular the influence of the environmental factors on the diversity and structure of the free-living Vibrio) and the authors make too broad conclusions regarding the evolutionary drivers for those two distinct communities based on little presented data.

Overall the methods are well-cited and based on established research, however organization, structure, and discussion of the results could be improved upon.

Primary Strengths

This manuscript parses and summarizes a large amount of data relevant for current and future characterizations of Vibrio communities. The figures represent the data and relationships the authors are trying to convey well. The motivation and reasoning for the direction of the statistical analyses and study of the Vibrio community structure is well described.

Primary Weaknesses

Some analyses between different communities are surface-level, although given that the paper's primary scope is within the context of Vibrio community dispersal, this "weakness" should not be held against the publication of the manuscript. The authors should take care not to over-interpret the more surface-level analyses. Some sections of the results are difficult to follow due to lack of topic organization and incomplete ideas, oftentimes the figure legends describe the results better than the main text.

Recommendations for improvement or requests for clarification are outlined below:

General suggestions:

- In context of the manuscript's results, "global" is a better word choice than "planetary"
- Formatting of section headers not consistent
- Majority of manuscript is very consistent in the use of "Vibrio", but there are several instances where Vibrio or vibrios is used instead

Introduction

Introduction clearly explains motivation. Hypothesis is clearly written.

Methods

- Overall well described, well-cited.
- Lines 368-371: fit better in the results section
- Lines 267-270: Authors state there is one prokaryotic enriched fraction (FLV), and four eukaryote-enriched fractions but proceed to list only three: NAV, MiAV, MeAV. Assumed they meant three eukaryote-enriched fractions for a total of four fractions, please correct.
- Line 271: Clarify language. Why are the living organisms found in the PAV fraction considered mainly eukaryotes versus a mix of eukaryote and prokaryote? Is this a result from sequencing bias or based on biomass?

Results & Figures

- Main text could use some editing for conciseness and clarity, particularly the "Vibrio and oceanic circulation" section
- The manuscript would benefit from a short paragraph at the start of the results section introducing the TARA Ocean expedition, number of stations, sampling times, etc. (similar to the beginning of the methods section) to provide context for the results
- Flow of the paper could be improved by either combining the "Bacterial community and Vibrio distribution in the oceans" section with the "vibrio species" section or move the "vibrio species" section from last to second in the Results

Lines 108-173: Vibrio and oceanic circulation section

- Break this section into multiple paragraphs based on topic
- In the Figure 3, NAV was changed to MAV. Please fix.
- Split Figure 3 into two figures, grouping 3A-D, and 3E-H. Corresponding break in the main text would be between 138 and 139.
- Provide citation(s) for statement in line 134-135
- Simplify any discussion of methods in the results section (notably lines 146-150)

Lines 146-149: there is no corresponding section in methods section discussing this topic. The manuscript would benefit from a more in-depth analysis and description of the influence environmental variables on the similarity of the Vibrio community between the different fractions based on region. As it is, the authors draw too strong and far-reaching conclusions in regards to local environmental adaptations and evolutionary drivers based on their singular result statement that "surface water temperature and salinity ... exerted a stronger influence on the FLV than the other fractions".

Figure legends

- Change "alfa" to "alpha"

Discussion

- Discussion of all results is lacking. Little effort was made to contextualize the results of this study to others regarding the composition of Vibrio species in different locations. Alternative explanations are not proposed
 - Authors did not revisit their hypothesis from the introduction
 - Discussion of future directions is limited and vague
 - Authors do not address any limitations/caveats to their approach. There is no discussion regarding potential sampling bias, effects of time of year/seasonality on their findings, or corrections for biomass differences in sample sizes
- Lines 191-222: The first paragraph is difficult to follow, ideas are often disjointed and not finished adequately. Flow is interrupted frequently. This section is not well cited in comparison to the rest of the manuscript.
- Replace "Microorganisms" with "microbial communities".
 - Do not use the word "probably"
- Line 240: Simplify or break-up the run-on sentence for clarity: "this recurrent admixture....might be....which may...."
- Line 245: Suggestion: change "revealed" to "emphasizes". This study is not the first to investigate plankton-associated Vibrio and plankton as a vector for Vibrio transport

(Remarks on code availability)

Code is provided for the statistical analyses and figures. While the code is not commented in detail, it is structured enough to follow-along. Yes, the provided code would be a useable resource for the community.

"does the code provide a README file with enough instructions for installing and running the application?" No, however there are many resources already available on how to install and run R. Only thing that could be clarified is the switch from Bash to to R.

Reviewer #2

(Remarks to the Author)

Doni et al. investigated global oceanic patterns of Vibrio communities in relation to ocean current travel time and geographical distance, using particle-size-fractionated and free-living water samples. Leveraging sequence data from the TARA Oceans expedition, the authors linked particle-associated fractions to plankton-associated microbial communities and observed that these communities exhibited genomic similarities across distant locations, particularly with higher correlations at ocean travel times of up to 1.5 years, mirroring known patterns of plankton dispersal. The authors further aligned their

findings with previous studies documenting the global spread of *Vibrio*-associated diseases, thereby corroborating their observed patterns and highlighting the public health implications of long-distance microbial dispersal in the ocean. Given that *Vibrio* species can be highly pathogenic and their ecological niches are expanding with climate change, this study provides important global insights into their spread and emergence. The manuscript is very well written, and I have only a few minor comments. I recommend this manuscript for publication in Nature Communications.

Comments

The authors provided an insightful discussion on the potential effects of plankton mediated selection on *Vibrio* species, while supporting their findings by showing that similarity was more strongly associated with travel time than with geographical distance. This may be beyond the scope of the present manuscript, but I would like to ask whether the authors have considered performing comparative genomic analyses on species that are differentially distributed between FLV and PAV. Such an analysis could help determine whether known physiological traits, based on available genome sequences, may explain the observed fractionation patterns driven by potential interactions.

LN 66: I think a PCoA plot would be more appropriate for Fig. 1A if the authors are referring to structural patterns. Alternatively, the authors could cite a different figure for this sentence and reserve Fig. 1A for the following sentence. LN 75: In Fig. B and C, do the panels represent frequency, meaning the proportion of samples with *Vibrio* detection? The text refers to "prevalence," and the figure caption uses "frequency," but the legend indicates that the color gradient represents abundance. Please clarify what metric is being shown.

Figure 1D caption: Typo. "alpha"

LN 95: In the main text I see the term NAV as well as other labels, but in the figures and methods I see MAV. I am not sure if I am misunderstanding something. Could the authors clarify the terminology?

LN 131-134: In Fig. 3D, does the y axis show the correlation coefficient? It is confusing because the axis label refers to similarities. Could the authors clarify what the metric represents?

(Remarks on code availability)

Version 1:

Reviewer comments:

Reviewer #1

(Remarks to the Author)

The authors have addressed all the comments made in the previous round of review. The revisions have significantly improved the clarity and quality of a great manuscript. The authors have neatly streamlined the revised results and discussion sections while maintaining the necessary context.

There is a minor typo in the legend of fig. S2. "alfa" -> "alpha"

The extra effort Doni et al. have invested in addressing the reviewer's comments is greatly appreciated and has substantially improved the manuscript, I recommend this manuscript for publication.

Reviewer #2

(Remarks to the Author)

The authors have adequately addressed the reviewer's comments, and I recommend publication of the manuscript.

Dear Editor,

We would like to thank you and the reviewers for the time dedicated to evaluating our manuscript and for providing valuable suggestions and comments, which we hope we have fully addressed in the revised version. Please find below our point-by-point responses to the reviewers' comments (bold text). References to revisions made in the manuscript (red text) are highlighted in yellow.

Reviewer #1

This manuscript describes the Vibrio community profile and dispersal across the oceans. In short, the authors describe the abundance of Vibrio in ocean surface waters, the deep chlorophyll maximum, and the mesopelagic, with an emphasis on the surface waters for more in-depth analyses. This study focused on broad trends and dispersal of Vibrio across the ocean based on previously established models. The authors highlight dispersal patterns amongst the plankton-associated Vibrio and oceanic currents, focusing on those areas with a transit time of 1.5 years. The key finding in this study highlights the plankton-associated Vibrio communities within different size fractions, and demonstrates clear evidence of the plankton size-dependent effect on the interconnectivity of Vibrio communities globally. An area in which this manuscript that could be improved upon is the discussion of the differences between the free-living Vibrio and the plankton-associated Vibrio fractions globally. The analysis between these two is surface-level (in particular the influence of the environmental factors on the diversity and structure of the free-living Vibrio) and the authors make too broad conclusions regarding the evolutionary drivers for those two distinct communities based on little presented data. Overall the methods are well-cited and based on established research, however organization, structure, and discussion of the results could be improved upon.

Primary Strengths

This manuscript parses and summarizes a large amount of data relevant for current and future characterizations of Vibrio communities. The figures represent the data and relationships the authors are trying to convey well. The motivation and reasoning for the direction of the statistical analyses and study of the Vibrio community structure is well described.

Primary Weaknesses

Some analyses between different communities are surface-level, although given that the paper's primary scope is within the context of Vibrio community dispersal, this "weakness" should not be held against the publication of the manuscript. The authors should take care not to over-interpret the more surface-level analyses. Some sections of the results are difficult to follow due to lack of topic organization and incomplete ideas, oftentimes the figure legends describe the results better than the main text.

The remark from the reviewers is correct. The central element of analysis in the study was on Vibrio, and that the taxonomic analysis of the whole bacterial community was only included to provide context for the Vibrio component (the 7 most abundant taxa). That is the reason of providing a very superficial description of the bacterial community found in the samples.

Recommendations for improvement or requests for clarification are outlined below:

General suggestions:

-In context of the manuscript's results, "global" is a better word choice than "planetary"

The term "Planetary" has been replaced with "global" consistently throughout the text.

-Formatting of section headers not consistent

Done

-Majority of manuscript is very consistent in the use of “Vibrio”, but there are several instances where Vibrio or vibrios is used instead

Now “Vibrio” is used consistently throughout the text.

Introduction

Introduction clearly explains motivation. Hypothesis is clearly written.

Methods

-Overall well described, well-cited.

-Lines 368-371: fit better in the results section

This paragraph was rephrased and moved to the results section (line 179).

-Lines 267-270: Authors state there is one prokaryotic enriched fraction (FLV), and four eukaryote-enriched fractions but proceed to list only three: NAV, MiAV, MeAV. Assumed they meant three eukaryote-enriched fractions for a total of four fractions, please correct.

This was a typographical error, as the PAV fractions are indeed three, which together with FLV result in a total of four fractions. This has now been corrected in the text (lines 291).

-Line 271: Clarify language. Why are the living organisms found in the PAV fraction considered mainly eukaryotes versus a mix of eukaryote and prokaryote? Is this a result from sequencing bias or based on biomass?

The reviewer’s remark is correct: we stated that the PAV bacterial fraction is a plankton-associated fraction that consists mainly of a mix of eukaryote and particle associated prokaryote. This sentence has now been clarified following the reviewer’s suggestions (lines 293-295).

Results & Figures

-Main text could use some editing for conciseness and clarity, particularly the “Vibrio and oceanic circulation” section

This section has been revised (lines 175–180), and split into subsections (lines 134 and 166) as suggested in a following comment, hopefully improving readability and clarity.

-The manuscript would benefit from a short paragraph at the start of the results section introducing the TARA Ocean expedition, number of stations, sampling times, etc. (similar to the beginning of the methods section) to provide context for the results.

Thanks for the suggestion, a section summarizing the relevant sampling data from the TARA Oceans Expedition has been added at the beginning of the Results section (lines 63-68), and a new figure showing the TARA stations and sampling locations has been included in the manuscript as Supplementary Figure (Fig. S1).

-Flow of the paper could be improved by either combining the “Bacterial community and Vibrio distribution in the oceans” section with the “vibrio species” section or move the “vibrio species” section from last to second in the Results

The “Vibrio species” section was moved from last to second in the results as suggested by the reviewer (lines 99-113).

Lines 108-173: *Vibrio* and oceanic circulation section

-Break this section into multiple paragraphs based on topic

This section was divided into two subsections “*Vibrio* transportation via ocean currents” (line 133) and “Connectivity of *Vibrio* communities at a global oceanic scale” (line 165) to improve the flow and highlight the main findings of the study.

-In the Figure 3, NAV was changed to MAV. Please fix.

Figure legend was updated accordingly

-Split Figure 3 into two figures, grouping 3A-D, and 3E-H. Corresponding break in the main text would be between 138 and 139.

Although we split the text into two subsections to improve flow and readability (see above), we prefer to keep Figure 3 as it is. Since panels A to H are highly interconnected, it is easier for the reader to follow and interpret the main results as described and discussed in the text. The break suggested by the reviewer would imply to display the maps in two different figures, which could make difficult to analyze and compare the results in a consistent way.

-Provide citation(s) for statement in line 134-135

New reference (Richter et al 2022, *Elife* 11, 1-30) has been incorporated in the text (line 161)

-Simplify any discussion of methods in the results section (notably lines 146-150)

The methods description in this section was streamlined for clarity (lines 174-178)

Lines 146-149: there is no corresponding section in methods section discussing this topic. The manuscript would benefit from a more in-depth analysis and description of the influence environmental variables on the similarity of the *Vibrio* community between the different fractions based on region. As it is, the authors draw too strong and far-reaching conclusions in regards to local environmental adaptations and evolutionary drivers based on their singular result statement that “surface water temperature and salinity ... exerted a stronger influence on the FLV than the other fractions”.

Methods describing this topic can be found at lines 380-384. We agree with the referee that no far-reaching conclusions can be drawn from the presented data. However, we believe it is beyond the scope of this paper to further investigate the influence of environmental variables, as this topic is already extensively covered in scientific literature. In addition, TARA sampling strategy does not cover a proper temporal scale and this may be a strong bias for this purpose. To address this, we have rephrased our statement in a more cautious manner (lines 176-178)

Figure legends

-Change “alfa” to “alpha”

Done (line 588)

Discussion

-Discussion of all results is lacking. Little effort was made to contextualize the results of this study to others regarding the composition of *Vibrio* species in different locations. Alternative explanations are not proposed

The discussion was entirely focused on contextualizing the results obtained toward a main goal: identifying broad trends and mechanisms of *Vibrio* dispersal across the global ocean. Taxonomic analysis of *Vibrio* communities was limited to identifying the main *Vibrio* species through co-

assembly of *Vibrio* reads across large oceanic regions and their association with different size fractions. Unfortunately, this approach constrained by the complex structure of the TARA marine metagenomes precludes a robust assessment of *Vibrio* biodiversity at local geographic scales and, consequently, a direct comparison with other studies addressing this topic. Another limitation was the lack of data and the absence of additional large-scale *Vibrio* studies in oceanic waters for comparison.

-Authors did not revisit their hypothesis from the introduction

A sentence was added, and the paragraph was rephrased to address this issue in the discussion (lines 234-237)

-Discussion of future directions is limited and vague

A paragraph emphasizing future directions has been added (lines 266-270)

-Authors do not address any limitations/caveats to their approach. There is no discussion regarding potential sampling bias, effects of time of year/seasonality on their findings, or corrections for biomass differences in sample sizes.

A paragraph addressing the potential caveats of the study has been added, particularly emphasizing the role of seasonality and interannual variation in shaping *Vibrio* communities, as this is certainly a major source of constraint in the TARA dataset (lines 262-265). This paragraph is directly linked to the future research directions outlined in our response to the previous comment.

-Lines 191-222: The first paragraph is difficult to follow, ideas are often disjointed and not finished adequately. Flow is interrupted frequently. This section is not well cited in comparison to the rest of the manuscript.

We made minor adjustments to the paragraph, hopefully improving its flow and readability (lines 203-215).

-Replace “Microorganisms” with “microbial communities”.

Done (line 210).

-Do not use the word “probably”

This sentence was rephrased (lines 207-209) and now reads: “FLV exhibited niche partitioning and a biogeographical structure, likely shaped by limited dispersal and isolation, which may contribute to ecological cohesion”

-Line 240: Simplify or break-up the run-on sentence for clarity: “this recurrent admixture....might be....which may....”

This sentence was rephrased (lines 254-257)

*-Line 245: Suggestion: change “revealed” to “emphasizes”. This study is not the first to investigate plankton-associated *Vibrio* and plankton as a vector for *Vibrio* transport*

Done (line 260).

Reviewer #1 (Remarks on code availability):

Code is provided for the statistical analyses and figures. While the code is not commented in detail, it is structured enough to follow-along. Yes, the provided code would be a useable resource for the community.

“does the code provide a README file with enough instructions for installing and running the

application?" No, however there are many resources already available on how to install and run R. Only thing that could be clarified is the switch from Bash to R.

We improved the readability of the code by specifying whether it was used in the Bash or R environment.

Reviewer #2 (Remarks to the Author):

Doni et al. investigated global oceanic patterns of Vibrio communities in relation to ocean current travel time and geographical distance, using particle-size-fractionated and free-living water samples. Leveraging sequence data from the TARA Oceans expedition, the authors linked particle-associated fractions to plankton-associated microbial communities and observed that these communities exhibited genomic similarities across distant locations, particularly with higher correlations at ocean travel times of up to 1.5 years, mirroring known patterns of plankton dispersal. The authors further aligned their findings with previous studies documenting the global spread of Vibrio-associated diseases, thereby corroborating their observed patterns and highlighting the public health implications of long-distance microbial dispersal in the ocean. Given that Vibrio species can be highly pathogenic and their ecological niches are expanding with climate change, this study provides important global insights into their spread and emergence. The manuscript is very well written, and I have only a few minor comments. I recommend this manuscript for publication in Nature Communications.

Comments

The authors provided an insightful discussion on the potential effects of plankton mediated selection on Vibrio species, while supporting their findings by showing that similarity was more strongly associated with travel time than with geographical distance. This may be beyond the scope of the present manuscript, but I would like to ask whether the authors have considered performing comparative genomic analyses on species that are differentially distributed between FLV and PAV. Such an analysis could help determine whether known physiological traits, based on available genome sequences, may explain the observed fractionation patterns driven by potential interactions.

This is an extremely interesting aspect that deserves closer investigation. Unfortunately, MAGs reconstruction, which would likely need to address these issues, could not be properly performed in this study for most Vibrio species. Our taxonomic analysis of Vibrio communities was therefore limited to identifying the main species through co-assembly of Vibrio reads into contigs across large oceanic regions and their association with different size fractions. Unfortunately, this approach, constrained by the complex structure of the TARA marine metagenomes, precludes robust comparative genomic analyses of Vibrio strains.

LN 66: I think a PCoA plot would be more appropriate for Fig. 1A if the authors are referring to structural patterns. Alternatively, the authors could cite a different figure for this sentence and reserve Fig. 1A for the following sentence.

The citation of Fig. 1A has been moved to the following sentences (lines 76 and 78)

LN 75: In Fig. B and C, do the panels represent frequency, meaning the proportion of samples with Vibrio detection? The text refers to "prevalence," and the figure caption uses "frequency," but the legend indicates that the color gradient represents abundance. Please clarify what metric is being shown.

Thanks. The metric used is "Relative abundance," which is now consistently applied throughout the text and figure captions (lines 81 and 586)

Figure 1D caption: Typo. "alpha"

done (line 588)

LN 95: In the main text I see the term NAV as well as other labels, but in the figures and methods I see MAV. I am not sure if I am misunderstanding something. Could the authors clarify the terminology?

This was a typo, the correct acronym is NAV (nanoplankton 5–20 μm). This has been corrected in the text (line 351)

LN 131-134: In Fig. 3D, does the y axis show the correlation coefficient? It is confusing because the axis label refers to similarities. Could the authors clarify what the metric represents?

Figure 3D (now 4D) does not show a correlation coefficient but rather compares Vibrio similarity values (y axis) for each fraction with travel time and distance in kilometers. Caption (line 614) and Text (lines 158-159) has been revised to avoid misunderstanding.